# Altered serum adiponectin and interleukin-8 levels are associated in the pathophysiology of major depressive disorder: A case-control study

Salsabil Islam⊗, Towhidul Islam📷 ⊗, Zabun Nahar, Mohammad Shahriar, Sardar Mohammad Ashraful Islam, Mohiuddin Ahmed Bhuiyan, Md. Rabiul Islam📷 *

Department of Pharmacy, University of Asia Pacific, Farmgate, Dhaka, Bangladesh

⊗ These authors contributed equally to this work.
* robi.ayaan@gmail.com

## Abstract

**Data Availability Statement:** All relevant data are within the manuscript.

### Background

Major depressive disorder (MDD) is a complex mental health condition that results in several obstacles including disabilities, loss of productivity, and economic burdens on both patients and society. Etiopathogenesis of MDD involves several factors such as sociodemographic, genetic, and biological determinants. However, any suitable biomarkers for risk assessment of depression have not been established yet. Alterations of cytokine are assumed to be involved in the pathophysiology and severity of the depressive disorder. Therefore, we aimed to evaluate serum adiponectin and interleukin-8 (IL-8) among MDD patients in Bangladesh.

### Methods

We recruited a total of 63 MDD patients and 94 age-sex matched healthy controls (HCs) in the present study. MDD patients were enrolled from a tertiary care teaching hospital, Dhaka, Bangladesh, and HCs from surrounding parts of Dhaka city. A psychiatrist assessed all the study participants following the criteria mentioned in the DSM-5. We applied the Hamilton depression (Ham-D) rating scale to assess the depression severity. Serum adiponectin and IL-8 levels were determined using ELISA kits (BosterBio, USA).

### Results

The mean serum concentration of adiponectin was decreased (30.67±4.43 μg/mL vs. 53.81±5.37 μg/mL), and the IL-8 level was increased (160.93±14.84 pg/mL vs. 88.68±6.33 pg/mL) in MDD patients compared to HCs. Sex-specific scatters plot graphs showed the distribution of adiponectin and IL-8 levels with Ham-D scores in MDD patients. Also, ROC curve analysis demonstrated good predictive performances of serum adiponectin and IL-8 for MDD with the area under the curve (AUC) as 0.895 and 0.806, respectively.

**Funding:** The author(s) received no specific funding for this work.

**Competing interests:** The authors have declared that no competing interests exist.

## Conclusion

The present study findings suggest that alterations of serum adiponectin and IL-8 levels in MDD patients might be involved in the disease process. Therefore, we can use these changes of cytokines in serum levels as early risk assessment tools for depression. The present study findings should be considered preliminary. We propose further interventional studies to evaluate the exact role of adiponectin and IL-8 in depression.

## Introduction

Major depressive disorder (MDD) is a complex and heterogeneous mental health disorder. Due to its high heterogeneity, many aspects of this mental disease are still unclear [1]. It is a prevalent disease that affects people who have other medical or neurologic disorders [2]. Social, environmental, psychological, genetic, and biological factors are involved in the etiopathogenesis of MDD [3–9]. However, the actual mechanism behind the etiopathogenesis and different subtypes and episodes of MDD is still to be understood [10]. Some researchers projected that MDD would be the global top-tier cause of disability by the upcoming decade [11]. MDD is a notable psychiatric disorder with a higher prevalence. At the same time, it is one of the most disabling disorders around the globe. MDD was ranked fourth among the disability-causing disorders (measured in disability-adjusted life years). This disorder also leads to a negative impact on public health. The effect of MDD is sometimes stronger than certain physical conditions such as diabetes mellitus, rheumatoid arthritis, and coronary heart disease [12]. Around 180 million people of all sexes and ages are suffering from depression [13]. According to a recent meta-analysis, the continent-wise prevalence of MDD is 16.7% in Asia, 11.5% in Africa, 11.9% in Europe, 13.4% in North America, 20.6% in South America, and 7.3% in Australia [14]. It is hard to precisely estimate the current circumstances in Bangladesh due to lacking data. Most Bangladeshi people underestimate mental disorders, particularly various types of depression [15]. About 16.05% of Bangladeshi adults have been suffering from psychiatric diseases [16]. However, MDD comprised 28.7% of all psychiatric disorders in Bangladesh [17]. Also, 4.4% of people of all ages and sexes here suffer from depression [18]. The prevalence of depression in other neighboring countries is India (3.9%), Pakistan (3.0%), Nepal (4%), and Bhutan (3.7%) [19].

Several biological mechanisms are involved in the pathogenesis of MDD [20,21]. Genetic polymorphism, monoamine deficiency, prolonged exposure to stress, hypothalamic-pituitary-adrenal axis activity imbalance, inflammatory cytokines alterations, and structural changes in the brain are prominent [22]. Fekadu et al. compiled and reported several factors for the etiology of depressive disorders [23]. The deficiency of monoamines such as serotonin and catecholamines causes various kinds of depression. Therefore, scientists target monoamine neurotransmitters to develop antidepressant drugs. Also, they discussed significant associations of neural circuitry, stress response circuits, inflammation, neuropeptides, hormones, circadian rhythm, and genetic and environmental factors with depression [23]. Most antidepressant drug classes such as SNRIs, [24] SSRIs, [25] NRIs, [26] MAOIs, [27] and TCAs [28] increase monoamine neurotransmitters in the brain. Therefore, scientists assumed that the pathogenesis of depression might be associated with altered cytokines, chemokines, BDNF, GDNF, neurotrophins, and growth factors in the human body [29]. However, the significance of certain relationships is negligible and remains inconsistent [30,31].

Over the last decade, several studies confirmed that physical and psychological stress leads to the progression and development of depressive disorders along with other mental diseases through several biochemical and hormonal mechanisms [22,23]. Some studies identified abnormal levels of ILs in the serum, especially cytokines [32]. Of the therapeutics, SSRIs and SNRIs are most frequently used in MDD [24–28]. These drugs possess significant but underestimated anti-inflammatory and anti-oxidative properties [33]. However, detailed mechanisms behind these actions have not been explored yet. The argument supports the basis of our study [33]. Nowadays, the role of inflammatory biomarkers in MDD progression and treatment response has been revealed extensively, and the investigation of peripheral biomarkers has become a common trend in research. According to a study, biomarkers such as CRP, TNF-α, IL-1β, IL-6, and BDNF were observed consistently in MDD and its treatment response [34]. As per a meta-analysis, there is an association between MDD and inflammatory biomarkers [35]. Another meta-analysis showed that higher levels of serum cytokines were observed in MDD patients than in healthy controls (HCs) [36]. However, we do not have any conclusive relationship has been found between cytokine levels in the peripheral blood and MDD. Some post-mortem studies reported higher TNF-α and IL-6 levels in CSF and TNF-α levels in the brain in MDD patients than in controls [37].

Association between adiponectin and depression has been demonstrated in many previous studies. Adipose tissue releases adiponectin into the blood circulation which provides anti-inflammatory and insulin-sensitizing effects [38]. Therefore, adiponectin levels are higher in plasma than other hormones [38,39]. Several studies explored that adiponectin levels would go down in people who were depressed, while others said that this cytokine would go up in people who had MDD [39]. Many researchers think that mood disorders are associated with insulin resistance and inflammation [38]. Anti-inflammatory cytokine adiponectin levels fall in people who are depressed because it is an anti-inflammatory cytokine [40]. A previous study reported adiponectin level was lower in MDD patients than in healthy people [41]. Scientists observe that there might have a negative relationship between the level of depression and the amount of adiponectin in the blood from several clinical studies. Also, there is a strong association between hypoadiponectinemia and depressive disorder. These findings suggest that lower serum adiponectin levels may be a risk factor for developing depression [41,42]. Also, adiponectin may play a vital role in the pathogenesis of depression [43]. As MDD is a mental disorder and adiponectin does several functions in the human brain, there can be a correlation between these two [44].

IL-8 is a neutrophil-activating cytokine that mainly exerts chemotactic activities; therefore, it is also called chemokine, CXC8 [45]. It is produced by different tissue and blood cells including fibroblasts and monocytes [45]. IL-8 has a particular target selectivity for neutrophils, with relatively minor effects on other blood cells [45,46]. Neutrophils are attracted to and activated by IL-8 in inflammatory areas, and some other cytokines' DNA sequence features with IL-8 suggest common regulatory pathways [46]. As MDD increases suicidal thoughts and suicidal behavior, studies of suicidal behavior might be useful to find a linkage between IL-8 and MDD [47]. Isung et al. studied 43 drug-free suicidal attempters and compared them with 20 HCs. They found for the first time that IL-8 levels were significantly lower among individuals with suicidal tendencies than in healthy volunteers [48]. However, Dahl et al. have argued these findings. Their cohort study has reported that serum IL-8 in MDD patients compared to HCs was significantly elevated at baseline. However, after 12 weeks of treatment, it was significantly decreased than baseline concentration [49]. Also, another study found that serum levels of IL-8 increased in depressive patients and decreased after 12 weeks of treatment [50]. Another study also reported significant elevation in MDD patients [51]. Moreover, some other studies also suggested similar findings [52].

The association between cytokine alterations and the pathogenesis of major depression is still obscure. Moreover, a balance between pro-inflammatory and anti-inflammatory effects is crucial in depression. Any imbalance in inflammatory markers might influence the pathophysiology of depression. IL-8 is prominently known for its pro-inflammatory effects and adiponectin provides anti-inflammatory effects. Based on the published evidence, we know little about the role of adiponectin and IL-8 in the development of depression. Therefore, the present study aimed to evaluate serum adiponectin and IL-8 in MDD through a case-control design.

## Materials and methods

### Study population

This prospective case-control study recruited 63 MDD patients and 94 HCs. We enrolled drug-free MDD patients from the psychiatry department of BSMMU (Bangabandhu Sheikh Mujib Medical University), Dhaka, Bangladesh. In Bangladesh, BSMMU has the honor of being the only medical college university. Therefore, it covers versatile patients from all over the country. Also, we enrolled age-sex matched HCs from different parts of Dhaka city. A qualified psychiatrist was involved in this inclusion process. The psychiatrist conducted the screening procedure by applying structured clinical interviews based on DSM-5 criteria. Also, we assessed the severity of MDD patients according to the Hamilton depression (Ham-D) rating scale. We recorded a detailed history of the study population, and their socio-demographic profiles using a structured questionnaire. We considered MDD when depressive symptoms lasted at least a couple of weeks. Our study population had an age range between 18 to 60 years old, had BMI within 16–34 kg/m$^2$, and a Hand-D score equal to or greater than seven. Participants with a history of comorbidity with other diseases such as immune, neural, infectious, diabetes mellitus and psychiatric diseases, history of alcoholism or other drug abuse, and abnormal BMI were excluded from this study.

### Blood sample collection and processing

Before the collection of the blood sample, participants were exposed to overnight fasting. Then, 5 ml of blood from each patient was collected from their cephalic vein from 8 to 9 am. After the collection of blood, we kept the samples in falcon tubes for clot formation within one hour at 25°C temperature. Then we separated serum samples from the blood sample using the centrifugation technique at 3000 rpm for 15 minutes. The extracted serum was aliquoted in polypropylene-made Eppendorf tubes and stored at -80°C for further analyses.

### Serum sample analysis

The serum levels of adiponectin and IL-8 were measured using ELISA techniques (Boater Bio, USA). We conducted the entire process according to instructions provided by the manufacturer. The assay sensitivity for adiponectin and IL-8 were <1 μg/mL and <60 pg/mL, respectively. Also, there was no cross-reaction with other cytokines. The same investigators performed the entire assay to avoid inter-assay variations, and the investigators were unaware of the clinical outcomes of the samples.

### Statistical analysis

We used Microsoft Excel for data processing. All statistical analyses were performed by IBM SPSS software package version 25.0 (IBM Corporation, Armonk, USA). We conducted descriptive statistics to check the data normality. Independent sample t-test, Fisher's exact test,

and Man-Whitney U-test were performed to compare the variables between the group. Box-plot graphs and scatters plot graphs were constructed to visualize the distribution of parameters between the groups. We performed receiver operating characteristic (ROC) curve analysis to evaluate the diagnostic performance analyzed parameters. All statistical analyses were considered significant at p-values less than or equal to 0.05.

## Ethics

We obtained ethical approval for this study from the Research Ethics Committee, University of Asia Pacific (Ref: UAP/REC/2021/101). The entire study was conducted following the Declaration of Helsinki. Before participation, we briefed them about the objective and purpose of this study to the participants. Also, we took written consent from each participant before engaging in this study. We obtained written consent from the legal guardians in the case where an individual's thinking capacity was suspected to be impaired.

## Results

### Characteristics of study participants

We presented the sociodemographic profile of study participants in Table 1. Among MDD patients, 50.79% were female, and 49.21% were male. In HCs, female and male proportions were 56.38% and 43.62%, respectively. The majority of study participants were 18–30 years of age (53.97%). The proportion of married participants (55.56%) was higher in patients than HCs (47.87%). A total of 55.56% of patients have completed their secondary level education, and the majority of HCs completed their graduation (55.32%) or other upper levels of studies (p = 0.014). A very high percentage of patients belonged to a low economic impression (93.65%) whereas, 55.32% of HCs belonged to a medium economic condition (p<0.001). Patients and controls of this study mostly lived in rural areas (55.56%) and urban areas (79.79%), respectively (p<0.001). This study's two groups' results on smoking habits are identical. Both groups in this study show majority of patients (79.37%) & the healthy population (77.66%) had no smoking habit (p = 0.799). 66.67% of patients had a previous history of MDD, and 73.02% did not have any family history of MDD.

### Clinical parameters and laboratory findings

The clinical profiles and laboratory results have been presented in Table 2. MDD patients showed lower serum adiponectin levels compared to HCs (30.67±4.43 μg/mL vs. 53.81±5.37 μg/mL; p = 0.002). Moreover, the changes in serum adiponectin were more prominent in females than males. However, we observed higher mean serum IL-8 concentrations in the MDD patients than HCs (160.93±14.84 pg/mL vs 88.68±6.33 pg/mL; p<0.001). The changes in serum adiponectin and IL-8 have been displayed in Fig 1. Both sexes of MDD patients have shown higher mean IL-8 concentration than HCs. Also, sex-specific scatter plot graphs showed altered serum adiponectin and IL-8 were associated with severity scores in MDD patients (Fig 2). Sex-specific scatters plot graphs showed that female MDD patients showed lower serum adiponectin and IL-8 levels at higher Ham-D scores, and male MDD patients showed opposite results.

### Receiver operating characteristic curve analysis

We performed ROC curve analysis for serum adiponectin and IL-8 among the study population (Fig 3). According to ROC curve analysis, serum adiponectin and IL-8 levels showed good predictive performances. The cut-off point of serum adiponectin and IL-8 were 20.88 μg/mL and 93.47 pg/mL, respectively. The area under the curve (AUC) was 0.895 and 0.806 for

**Table 1. Socio-demographic profile of the study population.**

| Characteristics | MDD patients (n = 63) Mean ± SEM | Healthy controls (n = 94) Mean ± SEM | *P value* |
|---|---|---|---|
| Age in years | 31.40 ± 1.18 | 31.54 ± 1.00 | 0.926 |
| 18–30 | 34 (53.97%) | 56 (59.57%) | |
| 31–45 | 24 (38.10%) | 25 (26.60%) | |
| 46–60 | 5 (7.94) | 13 (13.83%) | |
| Sex | | | 0.585 |
| Male | 31 (49.21%) | 41 (43.62%) | |
| Female | 32 (50.79%) | 53 (56.38%) | |
| Marital Status | | | 0.147 |
| Married | 35 (55.56%) | 45 (47.87%) | |
| Unmarried | 28 (44.44%) | 49 (52.13%) | |
| BMI (kg/m$^2$) | 23.31 ± 0.54 | 24.49 ± 0.33 | 0.068 |
| Below 18.5 (CED) | 4 (6.35) | 2 (2.13%) | |
| 18.5–25 (normal) | 39 (61.90%) | 54 (57.45%) | |
| Above 25 (obese) | 20 (31.75%) | 38 (40.43%) | |
| Education level | | | 0.014 |
| Illiterate | 1 (1.59%) | 2 (2.13%) | |
| Primary level | 8 (12.70%) | 5 (5.32%) | |
| Secondary level | 35 (55.56%) | 35 (37.23%) | |
| Graduate and above | 19 (30.16%) | 52 (55.32%) | |
| Occupation | | | < 0.001 |
| Business | 4 (6.35%) | 15 (15.96%) | |
| Service | 10 (15.87%) | 12 (12.77%) | |
| Unemployed | 8 (12.70%) | 2 (2.13%) | |
| Student | 0 (0.00%) | 27 (28.72%) | |
| Others | 41 (65.08%) | 38 (40.43%) | |
| Economic impression | | | < 0.001 |
| High | 0 (0.00%) | 21 (22.34%) | |
| Medium | 4 (6.35%) | 52 (55.32%) | |
| Low | 59 (93.65%) | 21 (22.34%) | |
| Smoking habit | | | 0.799 |
| Yes | 13 (20.63%) | 21 (22.34%) | |
| No | 50 (79.37%) | 73 (77.66%) | |
| Residence area | | | < 0.001 |
| Rural | 35 (55.56%) | 19 (20.21%) | |
| Urban | 28 (44.44%) | 75 (79.79%) | |
| Previous history of MDD | | | < 0.001 |
| Yes | 42 (66.67%) | 1 (1.06%) | |
| No | 21 (33.33%) | 93 (98.94%) | |
| Family history of MDD | | | < 0.001 |
| Yes | 17 (26.98%) | 2 (2.13%) | |
| No | 46 (73.02%) | 92 (97.87%) | |

Abbreviations: BMI, body mass index; CED, chronic energy deficiency; MDD, major depressive disorder; SEM, standard error mean.

adiponectin and IL-8, respectively. For adiponectin, lower values were assigned as the disease state, however, higher values were assigned as the disease state for IL-8. ROC curve showed sensitivity, specificity, positive predictive value (PPV), and negative predictive value (NPV) as

**Table 2. Clinical profile and laboratory findings of the study population.**

| Parameters | MDD patients (n = 63) Mean ± SEM | Healthy controls (n = 94) Mean ± SEM | p value |
|---|---|---|---|
| DSM-5 score | 6.84 ± 0.22 | 1.27 ± 0.15 | < 0.001 |
| Male (P/C:31/41) | 6.64 ± 0.32 | 1.02 ± 0.16 | < 0.001 |
| Female (P/C:32/53) | 7.03 ± 0.28 | 1.54 ± 0.26 | < 0.001 |
| Ham-D score | 15.39 ± 0.48 | 2.86 ± 0.35 | < 0.001 |
| Male (P/C:31/41) | 14.95 ± 0.73 | 2.30 ± 0.37 | < 0.001 |
| Female (P/C: 32/53) | 15.82 ± 0.64 | 3.47 ± 0.60 | < 0.001 |
| Serum adiponectin (μg/mL) | 30.67 ± 4.43 | 53.81 ± 5.37 | 0.002 |
| Male (P/C:31/41) | 27.72 ± 5.52 | 47.25 ± 8.24 | 0.071 |
| Female (P/C: 32/53) | 33.51 ± 6.94 | 58.89 ± 7.07 | 0.019 |
| BMI below 18.5 (P/C:3/2) | 13.20 ± 9.31 | 11.63 ± 7.42 | 0.945 |
| BMI 18.5–25 (P/C:39/54) | 29.49 ± 5.31 | 62.18 ± 8.11 | 0.003 |
| BMI above 25 (P/C:21/38) | 36.44 ± 9.17 | 43.31 ± 6.06 | 0.524 |
| Serum IL-8 (pg/mL) | 160.93 ± 14.84 | 88.68 ± 6.33 | < 0.001 |
| Male (P/C:31/41) | 166.21 ± 23.56 | 100.71 ± 12.88 | 0.011 |
| Female (P/C: 32/53) | 155.45 ± 18.22 | 79.65 ± 5.23 | < 0.001 |
| BMI below 18.5 (P/C:3/2) | 91.01 ± 29.43 | 30.57 ± 6.33 | 0.412 |
| BMI 18.5–25 (P/C:39/54) | 179.26 ± 20.49 | 77.85 ± 4.86 | < 0.001 |
| BMI above 25 (P/C:21/38) | 131.40 ± 16.04 | 69.48 ± 4.69 | 0.002 |

Abbreviations: BMI, body mass index; DSM-5, diagnostic and statistical manual for mental disorders, 5th edition; Ham-D, 17-item Hamilton depression rating scale; IL-8, interleukin-8; MDD, major depressive disorder; P/C, patients/control; SEM, standard error mean.

85.1%, 81.0%, 86.4%, and 88.5%, respectively for adiponectin and 73.0%, 71.3%, 76.5%, and 72.2%, respectively for IL-8 (Table 3).

## Discussion

The discovery and use of biomarkers for MDD are essential for the proper diagnosis and management of MDD patients. Adiponectin has a greater possibility to be considered as a potential

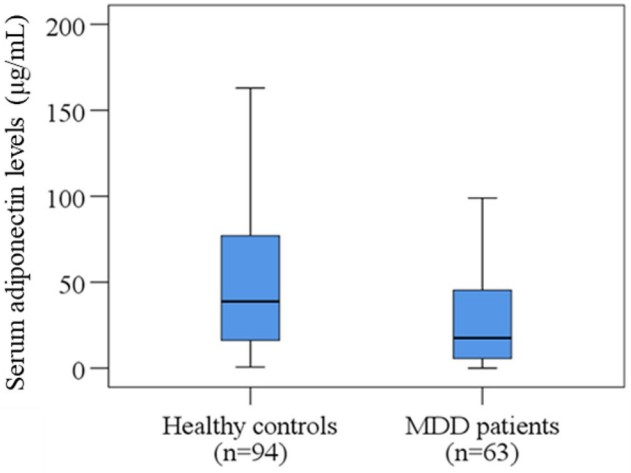
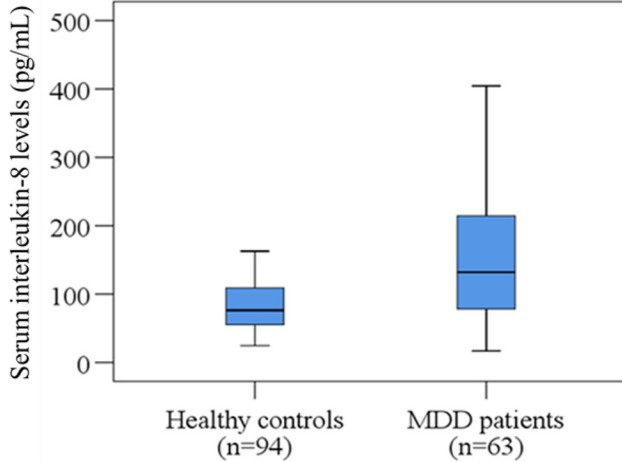

**Fig 1. Changes of serum adiponectin and interleukin-8 levels in MDD patients and healthy controls.** Boxplot graphs showing the median, maximum and minimum value range.

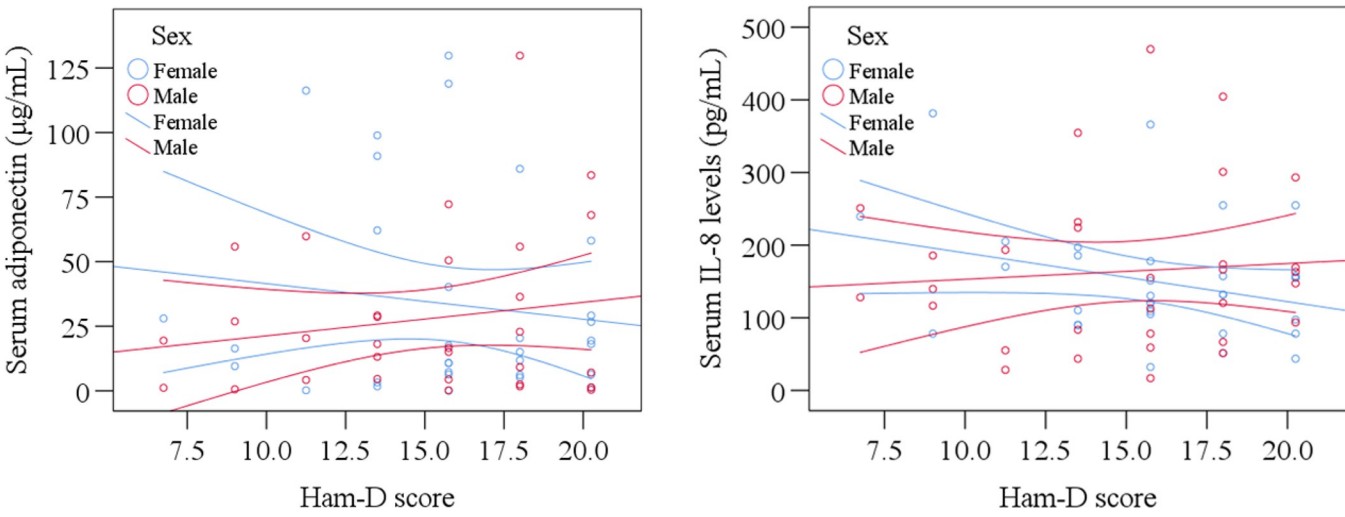

**Fig 2. Scatter plot graphs are showing sex-specific associations of altered adiponectin and interleukin-8 with severity scores in MDD patients.**

marker for depression. Because it is a polypeptide that controls glucose levels and the break-down of fatty acids [53]. Adiponectin has an antidepressant-like effect in social-defeat stress-induced depression in an animal model [54]. According to the present findings, MDD patients showed lower serum adiponectin levels than HCs. Also, few earlier studies found that the severity of MDD has been linked to a decreased adiponectin level [55,56]. Similarly, studies among elderly patients suffering from MDD also found a lower level of adiponectin compared to HCs [57]. However, BMI subgroups analysis showed that there was no dysregulation of serum adiponectin levels among the obese individuals.

Also, we observed a significant elevation of serum IL-8 levels in MDD patients compared to HCs. The findings of our study can be explained by various aspects of neuroinflammatory

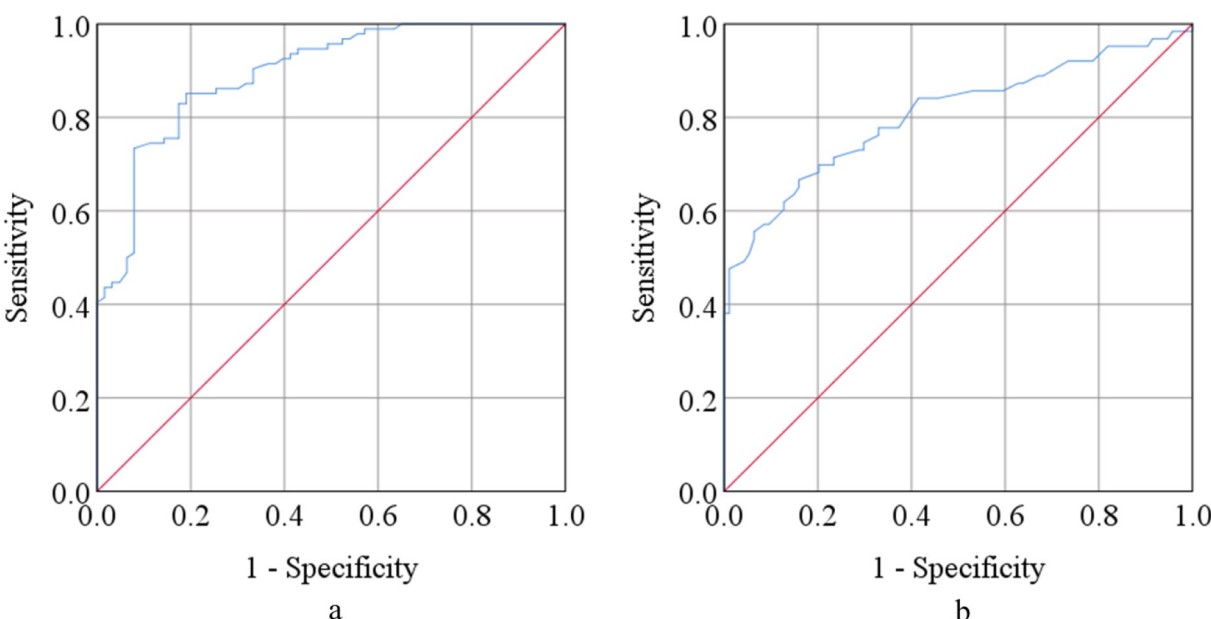

**Fig 3. Receiver operating characteristic (ROC) curves of serum adiponectin (a) and interleukin-8 (b) among the study population.**

**Table 3. Receiver operating characteristic curve analysis of serum adiponectin and interleukin-8.**

| Parameters | Cut-off value | Sensitivity (%) | Specificity (%) | PPV (%) | NPV (%) | AUC | 95% CI | | p-value |
|---|---|---|---|---|---|---|---|---|---|
| | | | | | | | Lower bound | Upper bound | |
| Adiponectin (μg/mL) | 20.88 | 85.1 | 81.0 | 86.4 | 88.5 | 0.895 | 0.846 | 0.944 | <0.001 |
| Interleukin-8 (pg/mL) | 93.47 | 73.0 | 71.3 | 76.5 | 72.2 | 0.806 | 0.731 | 0.881 | <0.001 |

Abbreviations: AUC, area under the curve; CI, confidence interval; PPV, positive predictive value; NPV, negative predictive value.

mechanisms [58]. There are numerous peripheral markers involved in the inflammatory responses to depression [59]. Etiopathogenesis of MDD involves several pathophysiological events. Among them, the inflammatory hypothesis and cytokine receptor activation of neurons are the most important mechanisms, for example, common pathways are activation of the HPA axis, [60] network dysfunction, reduced neuroplasticity, [61] kynurenine pathway activation associated with neuroprotective and neurodegenerative activity imbalance, [62] hyperexpression of serotonin transporters, [63] decreased neuronal growth factors, [64] neurotransmitter activities imbalance caused by altered production, release, and reuptake, [65] and obliviously neurodegeneration [66]. These mechanisms are linked with other social, physiological, and demographic factors [67]. In previous decades, several studies carried out on biomarkers covering a range of pro-and anti-inflammatory cytokines to establish the relationship between them with depressive disorders [35]. The association between MDD and the neurobiological basis of stress-induced changes to the brain and immune system supports these findings [58].

Moreover, a Korean group of researchers conducted a similar to ours. They reported a significant increase of IL-8 in serum among MDD patients than HCs. However, that elevation was not significantly correlated to disease severity [51]. Other similar studies also observed a significant elevation of serum IL-8 in MDD patients than HCs [52,68]. These studies are consistent with our findings. Therefore, the elevated serum IL-8 levels in MDD might be the result of disease and may contribute to the immune response by its pro-inflammatory chemotactic activity. IL-8 itself is a pro-inflammatory cytokine and has chemotaxis properties. Moreover, it also acts as an anti-inflammatory cytokine. The role of IL-8 depends on its concentration. Higher and lower concentrations show opposite directions in activities [69]. Moreover, several studies have been conducted regarding the evaluation of adiponectin and IL-8 in major depression, yet none of them completely explored the role of these markers in depression [51,52,55,56,68].

Elevation of peripheral IL-8 level occurs not only in MDD but also in many other clinical conditions associated with inflammation. As chronic inflammation is correlated with MDD, IL-8 is a pro-inflammatory biomarker, reducing IL-8 levels can be a helpful indicator in treatment. Elevation of this cytokine might be associated with genetic vulnerability for these clinical conditions [70]. Some researchers reported exercise decreases depression, psychological distress, tension, confusion, anger, and fatigue [71]. Findings of a recent study suggested the changes in cytokine levels followed by acute exercise, and IL-8 levels notably increase in intense exercise compared to moderate [72]. However, some researchers found elevated IL-8 levels following maximal exercise and exhaustive exercise such as marathon-running [73,74]. Sellami et al. found the cause of increasing IL-8 as skeletal muscle is a source of IL-8 during acute strength exercise [75]. Also, they reported a combination of flexibility, aerobic, balance, and strength exercises over a long-term period might reduce IL-8 levels [75]. Also, IL-8 promotes angiogenesis, which might contribute to how exercise improves conditions of mental health [76]. As adiponectin decreases in MDD patients compared to HCs, therefore, the

elevation of adiponectin levels in MDD patients might have a positive impact on treatment and management. However, decreased adiponectin levels also occurred in other diseases [77]. The study of adiponectin along with IL-8 is important when it is known that adiponectin is associated with the production of several inflammatory cytokines, and alterations in adiponectin can ultimately alter the levels of other cytokines such as IL-8 [78]. Since antidepressant drugs have anti-inflammatory activities. Therefore, anti-depressant therapy might restore the balance of serum adiponectin and IL-8 levels in MDD patients. Moreover, some specific endogenous body components have adiponectin increasing and IL-8 decreasing properties [79]. A previous study suggests that resistance training increases adiponectin levels in the human body [80]. This increased adiponectin might be responsible for different neuropsychological alterations. For example, the boosting effects on neurons in the human hippocampus help to alleviate an individual's depressive symptoms [81]. Therefore, we should focus on decreasing the MDD-mediated increased inflammation and oxidative stress by either exercise or anti-depressive therapy [82,83]; scientists also suggest probiotics improve the situation [84].

The present study performed ROC curve analysis for altered adiponectin and IL-8 to estimate the predictive performances (Fig 3). Precision based on the area under the curve (AUC) in ROC analysis are as follows: <0.6 = not useful, 0.6–0.7 = poor, 0.7–0.8 = fair, 0.8–0.9 = good, and 0.9–1.0 = excellent [85]. Many past studies recognized the altered serum adiponectin and IL-8 in depression. But these alterations cannot be used as risk assessment factors in depression due to the absence of their predictive performances [51,52,55,56,68,70]. According to the present study findings, the AUC of adiponectin and IL-8 were 0.895 and 0.806, respectively. Both the curve values were significant (p<0.001). Higher values of IL-8 and lower values of adiponectin were assigned as the disease state. Based on the ROC analysis results, both the markers showed good sensitivity, specificity, PPV, and NPV. Major depression multi-factorial disease where many environmental, biological, and genetic factors are involved [86]. Therefore, the identification and definition of biological risk factors are vital for understanding the pathophysiology and management of depression. Evidence suggests that recovered MDD patients achieved a balance in the levels of pro-and anti-inflammatory markers after treatment [87]. However, the previous studies could not suggest possible interventions to resume the balance of altered adiponectin and IL-8 in depression based on their findings [51,55,68,70]. The present study findings suggest some therapeutic approaches for MDD patients. Mind-body therapies (MBTs), physical exercises, and nutritional food/supplementation as non-pharmacological interventional strategies that can decrease stress levels, correct altered biomarkers, and improve the quality of life of MDD patients.

## Limitations of the present study

The present study has a few limitations. The case-control study design is a drawback, and we cannot find treatment response and alterations of peripheral adiponectin and IL-8 levels in MDD patients over time through this design. Also, measuring only adiponectin and IL-8 levels does not resemble the whole neuroinflammatory process of MDD. The measurement of other parameters in the same population and laboratory setup would be appropriate. We did not think about the effect of food habits on the analyzed parameters in this study. Therefore, recommend to conduct cohorts, RCTs, and longitudinal studies with multiple pro-and anti-inflammatory cytokines in a similar study population. Also, the present study findings would be better if we could provide biological subtypes of MDD, age of MDD onset, and the frequency and severity of depressive episodes. Alongside these limitations, our study has adequate strengths. For instance, this is the first-ever study in Bangladesh regarding the evaluation of adiponectin and IL-8. Globally, only a few studies have been conducted about this pro-

inflammatory cytokine, and some of them obtained inconsistent findings. The present study ensured diversification and homogeneity among the study population. Also, this study performed a sex-specific analysis of serum adiponectin and IL-8 levels in MDD patients.

## Practical implications

MDD is a complicated mental disorder that requires several approaches to manage. Because antidepressant medications alone are insufficient to treat moderate to severe depressive episodes, and because antidepressant medications have a limited therapeutic window, using solely antidepressant treatment is not recommended. As a result, a broad range of research data from different geographical locations assists clinicians in determining the severity of MDD and providing appropriate care. The outcomes of this research will help in the evaluation of depression risk. The actual associations of serum adiponectin and IL-8 with MDD have not been established yet. However, these results will contribute to the literature since they reflect the pathophysiology of MDD in a country having a large population. The findings of this research will aid in a better understanding of the pathophysiology of MDD. Since there are no risk assessment markers are not available for MDD at this time, these results may assist psychiatrists in gaining knowledge of how various factors influence the severity of MDD, which may aid them in a better understanding of MDD pathophysiology. Therefore, clinicians can use can use these altered markers as early risk assessment tools to evaluate depression risks.

## Conclusion

According to the present study findings, we can assume the associations between altered serum adiponectin and IL-8 levels and the pathophysiology of MDD. The good predictive performance of the ROC analysis gives confidence to the present study findings. Therefore, the altered serum adiponectin and IL-8 levels might be indicative to the development of depressive disorder. The current study findings should be considered preliminary, and we recommend further studies with large and more homogeneous samples to explore the role of the above markers on depression.

## Acknowledgments

We thank all the participants and their relatives for their cooperation to this study. Also, we would like to thank all physicians and administrative staffs at the department of psychiatry, BSMMU, for their cooperation and support to this study.

## Author Contributions

**Conceptualization:** Salsabil Islam, Towhidul Islam, Md. Rabiul Islam.

**Data curation:** Salsabil Islam, Towhidul Islam, Zabun Nahar.

**Formal analysis:** Salsabil Islam, Towhidul Islam, Zabun Nahar.

**Investigation:** Salsabil Islam, Towhidul Islam, Zabun Nahar, Md. Rabiul Islam.

**Methodology:** Mohammad Shahriar, Sardar Mohammad Ashraful Islam, Md. Rabiul Islam.

**Project administration:** Mohammad Shahriar, Sardar Mohammad Ashraful Islam.

**Supervision:** Mohiuddin Ahmed Bhuiyan, Md. Rabiul Islam.

**Writing – original draft:** Salsabil Islam, Towhidul Islam.

**Writing – review & editing:** Mohiuddin Ahmed Bhuiyan, Md. Rabiul Islam.

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
