## [Decision Letter · Decision Letter 0]

26 May 2022

PONE-D-21-41053Altered serum adiponectin and IL-8 levels are associated in the pathophysiology of major depressive disorder: A case-control studyPLOS ONE

Dear Dr. Rabiul Islam,

Thank you for submitting your manuscript to PLOS ONE. After careful consideration by two experts, we feel that it has merit but does not fully meet PLOS ONE’s publication criteria as it currently stands. Your manuscript requires extensive revision before a possible re-submission.

If you choose to submit a revised version, you are kindly requested to do so by July 12th. If you will need more time than this to complete your revisions, please reply to this message or contact the journal office at plosone@plos.org. Please include the following items when submitting your revised manuscript:A rebuttal letter that responds to each point raised by the academic editor and reviewer(s). You should upload this letter as a separate file labeled 'Response to Reviewers'.A marked-up copy of your manuscript that highlights changes made to the original version. You should upload this as a separate file labeled 'Revised Manuscript with Track Changes'.An unmarked version of your revised paper without tracked changes. You should upload this as a separate file labeled 'Manuscript'.

Since one of the experts recommanded rejection, please be aware that a revised version would be re-examined by the reviewers.

Best regards,

Sylvie Fournel-Gigleux

Academic Editor

PLOS ONE

Journal Requirements:

Additional Editor Comments :

The manuscript by Islam et al. describes alterations of serum adiponectin and IL-8 levels in MDD patients. They show a decrease in adiponectin level and an increase in IL-8 in MDD patients. A sex-specific analysis of serum adiponectin and IL-8 levels in MDD patients was also carried out.

Both reviewers find that the manuscript does not clearly state the purpose and impact of the study. In particular the relation between serum adiponectin and IL-8 levels in MDD patients should be explained.

Authors should clarify their conclusions and clearly explain what does their study bring over existing data.

Concerns raised by Reviewer 2 on Figures 2 and 3 have to be carefully addressed.

Reviewers' comments:

Reviewer's Responses to Questions

**Comments to the Author**

1. Is the manuscript technically sound, and do the data support the conclusions?

Reviewer #1: Yes

Reviewer #2: Partly

2. Has the statistical analysis been performed appropriately and rigorously? 

Reviewer #1: Yes

Reviewer #2: I Don't Know

3. Have the authors made all data underlying the findings in their manuscript fully available?

Reviewer #1: Yes

Reviewer #2: Yes

4. Is the manuscript presented in an intelligible fashion and written in standard English?

Reviewer #1: No

Reviewer #2: Yes

5. Review Comments to the Author

Reviewer #1: The manuscript by Islam et al describes alterations of serum adiponectin and IL-8 levels in

MDD patients. They report that adiponectin was decreased and IL-8 level was increased in MDD patients. Moreover, they perform sex-specific analysis of serum adiponectin and IL-8 levels in MDD patients.

Overall, I feel that this manuscript was interesting, but confusing in part because the writing was not clear. In its present form, the manuscript is extremely difficult to follow and this markedly diminishes the reader's interest, I would strongly recommend the following revisions:

1. Have the manuscript proof read by a native English speaker. Some parts of the manuscript are better written than others e.g. the discussion is poor compared to the rest of the manuscript.

2. What the relation between serum adiponectin and IL-8 levels in MDD patients.

Reviewer #2: The manuscript submitted by Islam et al. aimed to compare serum adiponectin and interleukin-8 (IL-8) levels between individuals with major depressive disorder (MDD) and healthy subjects. The data showed an elevated IL-8 concentration and a reduced adiponectin level in patient with MDD compared to healthy control, and an association between both serum cytokines concentrations and the disease severity. The study also indicated that serum adiponectin and IL-8 levels exhibited good predictive performances and so, can be used as risk assessment factors in depression.

My major criticism is that the goal of the study is not clearly described. Although the authors provide an exhaustive published literature on the topic, they failed to address the research question. More especially, they mentioned contradictory findings in some previous studies. Some of them have shown differences in serum adiponectin and IL-8 between individuals with MDD and healthy subjects while others did not. They also claimed that the role of adiponectin and IL-8 in the development of depression remains unclear and indicated that any suitable biomarkers for risk assessment of depression have not been established yet. So, what is the aim of the adiponectin and IL-8 assessment? Did the authors want to explain the contradictory findings, or investigate the role of these cytokines in depression or identify new biomarkers for MDD? The question posed should be more defined.

Other comments

Originality. The authors did not state the novelty of the study and the manuscript provides minor advance over existing information.

Introduction. The authors provide numerous previous published data to introduce the topic of the study. However, the introduction section includes some details that are not required. For instance, I think that too much information on MDD prevalence and on pro-inflammatory changes in individuals with MDD are provided. Conversely, the contradictory findings on serum adiponectin and IL-8 should be described to better relate the study to previously published literature.

Methods. Overall, the research is technically sound. However, clinical characteristics that might represent biological subtypes of MDD and so might may influence the data, are lacking. The age of MDD onset, the speed of onset of depressive episodes or the medication status of the patients should be included. In addition, the authors claimed that participants with some criteria were excluded from the study. But what about patients with diabetes mellitus? This is an important point as this metabolic disorder may affect serum adiponectin levels. The ethics statement should be added in the method section.

Results.

The data are, for the most part, well set out. Figures 2 and 3 raise however some concerns:

- Figure 2. I do not understand the scatters plot graphs. What is this representation? According to the authors, Spearman’s correlation analysis have been performed to evaluate association among the parameters. If true, r (the coefficient correlation) and p (the statistical significance) values should be provided. What are the y values shown on graphs? r value should be between -1 and 1.

- Figure 3. I am not aware of the ROC curve analysis. I think that the findings would be more convincing if the authors explain both the significance of such analysis and how the conclusion may contribute to the study.

Other points:

As adipokines dysregulation is found in obese individuals, it would be more appropriate to consider the 3 BMI subgroups to compare serum adiponectin levels between MDD patients and healthy control. A meta-analysis suggested actually that studies involving samples with BMI ≥ 25 had lower adiponectin levels in subjects with MDD compared to HC (Cao et al., J Affect Disord, 2018).

The authors claimed that “the changes in serum adiponectin were prominent in females than in males” (p8). The data in table 2 do not support this interpretation. The mean serum adiponectin level was reduced by 43% in female MDD patient and by 41% in male patient.

Please check for the unit used for serum IL-8. Usually, circulating cytokine levels are not so elevated.

Conclusion

The authors should discuss the contribution of their findings to the previous published data. More especially, they should state whether their results might explain the contradictory studies.

Why the authors have considered only physical activity as a way to regulate serum cytokine levels while it is likely that antidepressant treatment might also change cytokines production?

The authors’ conclusion that “alterations of serum adiponectin and IL-8 may be the result of the pathophysiology of depression, and not the cause” is not supported by the data. Similarly, the data do not provide enough evidence to suggest the practical implications stated by the authors.

6. PLOS authors have the option to publish the peer review history of their article (what does this mean?). If published, this will include your full peer review and any attached files.

Reviewer #1: No

Reviewer #2: No

---

## [Author Response · Author response to Decision Letter 0]

1 Jul 2022

Dear Editors and Reviewers,

Thank you for your letter and the reviewers' comments on our manuscript entitled "Altered serum adiponectin and IL-8 levels are associated in the pathophysiology of major depressive disorder: A case-control study" (Manuscript ID PONE-D-21-41053). All the comments were valuable and helpful to the revision and improvement of the manuscript. We have carefully studied the comments and made corrections, which we hope will merit your approval. We marked the revised portions using track changes. Our point-by-point answers to the reviewers’ comments appear at the end of this letter.

We earnestly appreciate the Editors'/Reviewers' work. We hope that after this revision, the paper will be deemed fit for publication. We would be glad to respond to any further questions and comments that you may have. 

Once again, thank you very much for your comments and suggestions.

Best regards,

Md. Rabiul Islam, PhD

Associate Professor, Department of Pharmacy, University of Asia Pacific

74/A Green Road, Farmgate, Dhaka-1205, Bangladesh. 

Email: robi.ayaan@gmail.com; Cell: +8801916031831; ORCID iD: https://orcid.org/0000-0003-2820-3144

Point by point authors’ responses to the reviewers

Manuscript ID PONE-D-21-41053

Title: Altered serum adiponectin and IL-8 levels are associated in the pathophysiology of major depressive disorder: A case-control study

Reviewer #1

The manuscript by Islam et al describes alterations of serum adiponectin and IL-8 levels in

MDD patients. They report that adiponectin was decreased and IL-8 level was increased in MDD patients. Moreover, they perform sex-specific analysis of serum adiponectin and IL-8 levels in MDD patients.

Overall, I feel that this manuscript was interesting, but confusing in part because the writing was not clear. In its present form, the manuscript is extremely difficult to follow and this markedly diminishes the reader's interest, I would strongly recommend the following revisions:

Author responses

Thank you for your review and valuable observation. We appreciate your encouraging comments on our manuscript. The English language editing has been completed.

Comment 1:

Have the manuscript proof read by a native English speaker. Some parts of the manuscript are better written than others e.g., the discussion is poor compared to the rest of the manuscript.

Author responses

We have now edited the whole manuscript for correcting mistakes in both grammar and style by the person who is fluent in English. We think the revised manuscript is much better than previous one in terms of written English.

Comment 2:

What the relation between serum adiponectin and IL-8 levels in MDD patients.

Author responses

Thank you for your observation. We have now provided justification for combined assessment of adiponectin and IL-8 in depression in the revised manuscript. You will find this information in page 5, line 32-33; page 6, line 1-6.

The association between cytokine alterations and the pathogenesis of major depression is still obscure. Moreover, a balance between pro-inflammatory and anti-inflammatory effects is crucial in depression. Any imbalance in inflammatory markers might influence the pathophysiology of depression. IL-8 is prominently known for its pro-inflammatory effects and adiponectin provides anti-inflammatory effects. Based on the published evidence, we know little about the role of adiponectin and IL-8 in the development of depression. Therefore, the present study aimed to evaluate serum adiponectin and IL-8 in MDD through a case-control design.

Reviewer #2

The manuscript submitted by Islam et al. aimed to compare serum adiponectin and interleukin-8 (IL-8) levels between individuals with major depressive disorder (MDD) and healthy subjects. The data showed an elevated IL-8 concentration and a reduced adiponectin level in patient with MDD compared to healthy control, and an association between both serum cytokines concentrations and the disease severity. The study also indicated that serum adiponectin and IL-8 levels exhibited good predictive performances and so, can be used as risk assessment factors in depression.

Author responses

Thank you for your effort to review our manuscript.

Comment 1:

My major criticism is that the goal of the study is not clearly described. Although the authors provide an exhaustive published literature on the topic, they failed to address the research question. More especially, they mentioned contradictory findings in some previous studies. Some of them have shown differences in serum adiponectin and IL-8 between individuals with MDD and healthy subjects while others did not. They also claimed that the role of adiponectin and IL-8 in the development of depression remains unclear and indicated that any suitable biomarkers for risk assessment of depression have not been established yet. So, what is the aim of the adiponectin and IL-8 assessment? Did the authors want to explain the contradictory findings, or investigate the role of these cytokines in depression or identify new biomarkers for MDD? The question posed should be more defined.

Author responses

Thank you for your observation. We have now provided justification for combined assessment of adiponectin and IL-8 in depression in the revised manuscript. You will find this information in page 5, line 32-33; page 6, line 1-6.

The association between cytokine alterations and the pathogenesis of major depression is still obscure. Moreover, a balance between pro-inflammatory and anti-inflammatory effects is crucial in depression. Any imbalance in inflammatory markers might influence the pathophysiology of depression. IL-8 is prominently known for its pro-inflammatory effects and adiponectin provides anti-inflammatory effects. Based on the published evidence, we know little about the role of adiponectin and IL-8 in the development of depression. Therefore, the present study aimed to evaluate serum adiponectin and IL-8 in MDD through a case-control design.

Comment 2:

Other comments

Originality. The authors did not state the novelty of the study and the manuscript provides minor advance over existing information.

Author responses

Thank you for your opinion. We have now added some information based on what we already know about this topic and what is the present research’s contribution to the field. You will find this information in page 11, line 30-33; page 12, line 1-16.

The present study performed ROC curve analysis for altered adiponectin and IL-8 to estimate the predictive performances (Figure 3). Precision based on the area under the curve (AUC) in ROC analysis are as follows: <0.6 = not useful, 0.6-0.7 = poor, 0.7-0.8 = fair, 0.8-0.9 = good, and 0.9-1.0 = excellent [85]. Many past studies recognized the altered serum adiponectin and IL-8 in depression. But these alterations cannot be used as risk assessment fac-tors in depression due to the absence of their predictive performances [51,52,55,56,68,70]. According to the present study findings, the AUC of adiponectin and IL-8 were 0.895 and 0.806, respectively. Both the curve values were significant (p<0.001). Higher values of IL-8 and lower values of adiponectin were assigned as the disease state. Based on the ROC analysis results, both the markers showed good sensitivity, specificity, PPV, and NPV. Major de-pression multifactorial disease where many environmental, biological, and genetic factors are involved [86]. Therefore, the identification and definition of biological risk factors are vital for understanding the pathophysiology and management of depression. Evidence suggests that recovered MDD patients achieved a balance in the levels of pro-and anti-inflammatory markers after treatment [87]. However, the previous studies could not suggest possible interventions to resume the balance of altered adiponectin and IL-8 in depression based on their findings [51,55,68,70]. The present study findings suggest some therapeutic approaches for MDD patients. Mind-body therapies (MBTs), physical exercises, and nutritional food/supplementation as non-pharmacological interventional strategies that can decrease stress levels, correct altered biomarkers, and improve the quality of life of MDD patients.

Comment 3:

Introduction. The authors provide numerous previous published data to introduce the topic of the study. However, the introduction section includes some details that are not required. For instance, I think that too much information on MDD prevalence and on pro-inflammatory changes in individuals with MDD are provided. Conversely, the contradictory findings on serum adiponectin and IL-8 should be described to better relate the study to previously published literature.

Author responses

Thank you for your valuable suggestion. We have now revised the whole introduction section and reduced some less important information to make it more related to the present topic. 

Comment 4:

Methods. Overall, the research is technically sound. However, clinical characteristics that might represent biological subtypes of MDD and so might may influence the data, are lacking. The age of MDD onset, the speed of onset of depressive episodes or the medication status of the patients should be included. In addition, the authors claimed that participants with some criteria were excluded from the study. But what about patients with diabetes mellitus? This is an important point as this metabolic disorder may affect serum adiponectin levels. The ethics statement should be added in the method section.

Author responses

Thank you for your suggestion. 

It would be better if we could provide biological subtypes of MDD, age of MDD onset, the speed of onset of depressive episodes in the present article. We have added these as limitations of the present study (page 12, line 26-28).

We enrolled drug-free MDD patients for this study and excluded patients with diabetes mellitus. This information we mentioned in the method section (page 6, line 10, 22). 

The ethics statement has been added in the last paragraph of method section (page 7, line 20-25).

Comment 5:

Results.

The data are, for the most part, well set out. Figures 2 and 3 raise however some concerns:

- Figure 2. I do not understand the scatters plot graphs. What is this representation? According to the authors, Spearman’s correlation analysis have been performed to evaluate association among the parameters. If true, r (the coefficient correlation) and p (the statistical significance) values should be provided. What are the y values shown on graphs? r value should be between -1 and 1.

- Figure 3. I am not aware of the ROC curve analysis. I think that the findings would be more convincing if the authors explain both the significance of such analysis and how the conclusion may contribute to the study.

Author responses

Thank you for your opinion. 

We have now revised the Figure 2 by changing dimensions (x and y axes). The mean values in the fit lines are showing how adiponectin and IL-8 levels are changing with Ham-D score of MDD patients. We are seeing sex-specific changes through Figure 2 where we see female MDD patients showed lower serum levels at higher Ham-D scores and vice versa. Moreover, the rate of sex-specific changes of serum markers among MDD patients are higher among females than males.

We have now reported the r and p values of Spearman’s correlation analysis in the revised manuscript (page 8, line 21-26).

The y values in the earlier figure 2 were mean Ham-D scores of MDD patients by sex. These values are repetitive of Table 2, therefore, we omitted them from in the revised Figure 2. 

We have now explained how the findings from ROC would contribute to the earlier findings and based on these findings how researchers can use present findings for identification of risk assessment markers for depression (page 9, line 5-13; page 11, line 30-33; page 12, line 1-16).

Comment 6:

Other points:

As adipokines dysregulation is found in obese individuals, it would be more appropriate to consider the 3 BMI subgroups to compare serum adiponectin levels between MDD patients and healthy control. A meta-analysis suggested actually that studies involving samples with BMI ≥ 25 had lower adiponectin levels in subjects with MDD compared to HC (Cao et al., J Affect Disord, 2018).

Author responses

Thank you for your valuable suggestion. We have now analyzed and presented serum markers among 3 BMI subgroups (Revised Table 2). However, we have seen that no changes of adiponectin levels among the samples with BMI ≥ 25 in subjects with MDD compared to HC.

Comment 7:

The authors claimed that “the changes in serum adiponectin were prominent in females than in males” (p8). The data in table 2 do not support this interpretation. The mean serum adiponectin level was reduced by 43% in female MDD patient and by 41% in male patient.

Author responses

Thank you for your opinion. For serum adiponectin levels were 27.72 ± 5.52 in male MDD patients and 47.25 ± 8.24 in male HCs (p=0.071), however, the levels were 33.51 ± 6.94 female MDD patients and 58.89 ± 7.07 in female HCs (p=0.019) (Table 2). From this data we can see that female participants showed significant changes whereas males do not. Also, you will see the sharp changes of serum adiponectin levels among females MDD patients than males in the revised Figure 2.

Comment 8:

Please check for the unit used for serum IL-8. Usually, circulating cytokine levels are not so elevated.

Author responses

Thank you for your observation. We have checked the unit of serum IL-8 and found same.

Comment 9:

Conclusion

The authors should discuss the contribution of their findings to the previous published data. More especially, they should state whether their results might explain the contradictory studies.

Why the authors have considered only physical activity as a way to regulate serum cytokine levels while it is likely that antidepressant treatment might also change cytokines production?

The authors’ conclusion that “alterations of serum adiponectin and IL-8 may be the result of the pathophysiology of depression, and not the cause” is not supported by the data. Similarly, the data do not provide enough evidence to suggest the practical implications stated by the authors.

Author responses 

Thank you for your observation. We have now revised the conclusion. 

Additional Editor Comments:

The manuscript by Islam et al. describes alterations of serum adiponectin and IL-8 levels in MDD patients. They show a decrease in adiponectin level and an increase in IL-8 in MDD patients. A sex-specific analysis of serum adiponectin and IL-8 levels in MDD patients was also carried out.

Both reviewers find that the manuscript does not clearly state the purpose and impact of the study. In particular the relation between serum adiponectin and IL-8 levels in MDD patients should be explained.

Authors should clarify their conclusions and clearly explain what does their study bring over existing data.

Concerns raised by Reviewer 2 on Figures 2 and 3 have to be carefully addressed.

Author responses 

Thank you for your suggestion. We have revised the manuscript carefully following both reviewer’s comments and suggestions.

---

## [Decision Letter · Decision Letter 1]

26 Aug 2022

PONE-D-21-41053R1Altered serum adiponectin and interleukin-8 levels are associated in the pathophysiology of major depressive disorder: A case-control studyPLOS ONE

Dear Dr. Rabiul Islam

Thank you for submitting your manuscript to PLOS ONE. After careful consideration, we feel that your manuscript has significantly improved and can be suitable for publication once you have carefully carrefully carried out the revisions suggested by reviewer 2. We invite you to submit a revised version of the manuscript that addresses the points raised by this reviewer.

We look forward to receiving your revised manuscript.

Kind regards,

Sylvie Fournel-Gigleux

Academic Editor

PLOS ONE

Journal Requirements:

Additional Editor Comments :

This manuscript is suitable for publication once revisions suggested by reviewer 2 have been performed.

Reviewers' comments:

Reviewer's Responses to Questions

**Comments to the Author**

1. If the authors have adequately addressed your comments raised in a previous round of review and you feel that this manuscript is now acceptable for publication, you may indicate that here to bypass the “Comments to the Author” section, enter your conflict of interest statement in the “Confidential to Editor” section, and submit your "Accept" recommendation.

Reviewer #1: All comments have been addressed

Reviewer #2: Minor revision

2. Is the manuscript technically sound, and do the data support the conclusions?

Reviewer #1: Yes

Reviewer #2: Yes

3. Has the statistical analysis been performed appropriately and rigorously? 

Reviewer #1: Yes

Reviewer #2: Yes

4. Have the authors made all data underlying the findings in their manuscript fully available?

Reviewer #1: Yes

Reviewer #2: Yes

5. Is the manuscript presented in an intelligible fashion and written in standard English?

Reviewer #1: Yes

Reviewer #2: Yes

6. Review Comments to the Author

Reviewer #1: In this revision, the authors had addressed majority of my comments and I don't have any additional comments beyond what I raised in the initial review.

Reviewer #2: The reviewer would like to thank the authors for taking the recommendations into consideration and for their extensive point-by-point response to the initial review. Many efforts have been made to complete the paper with the clarifications and additional text. Overall, this is a well-written and thoughful paper whose findings clearly address an important clinical issue for MDD patients. With that said, there are still minor issues that the authors should consider:

• Thank you for adding data from Spearman’s correlation analysis. However, they do not provide significant conclusions and the interpretation is not correct. A so low r value found for serum adiponectin levels means that this cytokine may not be associated with MDD. In addition, the authors claimed that IL-8 levels showed opposite results compared to adiponectin whereas both r value are minus number. I think that Spearman’s correlation analysis should be discarded because they provided confusing findings over data from scatter plot graphs (figure 2).

• Although the authors have checked the unit used for serum IL-8 levels, I doubt whether they reach so high values. I did not get any information on the ELISA assay kit (I can’t get hold of the manufacturer anywhere), but serum IL-8 levels were shown to be ranged from 1 to 1000 pg/ml (An Z et al. Cell cycle, 2019, vol.18, p2928; - Amoras EDSG et al. Biomolecules, 2021 vol. 11 p1664 – Heldt S et al. Mycoses, 2017, vol.60, p818 – Sinegar GA et al. Acta Med Indones, 2015, vol.47, p120). Serum IL-8 levels around µg/ml are therefore unlikely.

• Some affirmations are not correct. At the beginning of the paragraph on the “Potential limitations of the study”, the authors cannot claim that “their approach aimed to be comprehensive” as it is a descriptive study. Similarly, they cannot state that “the altered serum adiponectin and IL-8 levels can be used as early risk assessment tools for depression” because the age of MDD onset remains unknown.

7. PLOS authors have the option to publish the peer review history of their article (what does this mean?). If published, this will include your full peer review and any attached files.

Reviewer #1: No

Reviewer #2: No

---

## [Author Response · Author response to Decision Letter 1]

8 Sep 2022

Dear Editors and Reviewers,

Thank you again for your letter and the reviewers' comments on our manuscript entitled "Altered serum adiponectin and IL-8 levels are associated in the pathophysiology of major depressive disorder: A case-control study" (Manuscript ID PONE-D-21-41053.R1). All the comments were valuable and helpful to the revision and improvement of the manuscript. We have carefully studied the comments and made corrections, which we hope will merit your approval. We marked the revised portions using track changes. Our point-by-point answers to the reviewers’ comments appear at the end of this letter.

We earnestly appreciate the Editors'/Reviewers' work. We hope that after this revision, the paper will be deemed fit for publication. We would be glad to respond to any further questions and comments that you may have. 

Once again, thank you very much for your comments and suggestions.

Best regards,

Md. Rabiul Islam, PhD

Associate Professor, Department of Pharmacy, University of Asia Pacific

74/A Green Road, Farmgate, Dhaka-1205, Bangladesh. 

Email: robi.ayaan@gmail.com; Cell: +8801916031831; ORCID iD: https://orcid.org/0000-0003-2820-3144

Point by point authors’ responses to the reviewers

Manuscript ID PONE-D-21-41053.R1

Title: Altered serum adiponectin and IL-8 levels are associated in the pathophysiology of major depressive disorder: A case-control study

Reviewer #1

In this revision, the authors had addressed majority of my comments and I don't have any additional comments beyond what I raised in the initial review.

Author responses

Thank you for your review and valuable opinion on the revised version of our manuscript.

Reviewer #2

The reviewer would like to thank the authors for taking the recommendations into consideration and for their extensive point-by-point response to the initial review. Many efforts have been made to complete the paper with the clarifications and additional text. Overall, this is a well-written and thoughtful paper whose findings clearly address an important clinical issue for MDD patients. With that said, there are still minor issues that the authors should consider:

Author responses

Thank you for your effort to review our manuscript and encouraging comments.

Comment 1:

• Thank you for adding data from Spearman’s correlation analysis. However, they do not provide significant conclusions and the interpretation is not correct. A so low r value found for serum adiponectin levels means that this cytokine may not be associated with MDD. In addition, the authors claimed that IL-8 levels showed opposite results compared to adiponectin whereas both r value are minus number. I think that Spearman’s correlation analysis should be discarded because they provided confusing findings over data from scatter plot graphs (figure 2).

Author responses

Thank you for your observation and valuable suggestion. We also believe that Spearman’s correlation analysis does not add significant value to the present findings, at the same time creating some confusion. Therefore, we discarded this analysis from the method, results and discussion sections following the reviewer’s suggestion. 

Comment 2:

• Although the authors have checked the unit used for serum IL-8 levels, I doubt whether they reach so high values. I did not get any information on the ELISA assay kit (I can’t get hold of the manufacturer anywhere), but serum IL-8 levels were shown to be ranged from 1 to 1000 pg/ml (An Z et al. Cell cycle, 2019, vol.18, p2928; - Amoras EDSG et al. Biomolecules, 2021 vol. 11 p1664 – Heldt S et al. Mycoses, 2017, vol.60, p818 – Sinegar GA et al. Acta Med Indones, 2015, vol.47, p120). Serum IL-8 levels around µg/ml are therefore unlikely.

Author responses

Thank you for your valuable observation. The unit of our calculated serum adiponectin and IL-8 levels was also pg/ml (Ref: https://www.bosterbio.com/datasheet?sku=EK0413;
https://www.bosterbio.com/datasheet?sku=EK0595). We wrote the unit as μg/ml by mistake for both the markers and we have now corrected the unit throughout the manuscript and other places where applicable (manuscript, Table 2, Figure 1 & 2). We are showing our apology for this mistake. 

Comment 3:

• Some affirmations are not correct. At the beginning of the paragraph on the “Potential limitations of the study”, the authors cannot claim that “their approach aimed to be comprehensive” as it is a descriptive study. Similarly, they cannot state that “the altered serum adiponectin and IL-8 levels can be used as early risk assessment tools for depression” because the age of MDD onset remains unknown.

Author responses

Thank you for your valuable suggestion. We have now corrected them accordingly (page 12, line 5-6; page 13, lines 10-11).

---

## [Decision Letter · Decision Letter 2]

11 Oct 2022

Altered serum adiponectin and interleukin-8 levels are associated in the pathophysiology of major depressive disorder: A case-control study

PONE-D-21-41053R2

Dear Rabiul Islam,

We’re pleased to inform you that your manuscript has been judged scientifically suitable for publication **once you have made the ****necessary corrections**
**to the serum adiponectin serum units as indicated by reviewer 2.**

Kind regards,

Sylvie Fournel-Gigleux

Academic Editor

PLOS ONE

Additional Editor Comments (optional):

The editor appreciate the effort made by the authors in addressing the previous comments of Reviewer 2. However, there is one mistake that the authors should correct. Although issues concerning IL-8 serum levels were properly addressed, we do not understand why the authors have also changed without apparent reason the unit for serum adiponectin levels. Adiponectin has a physiological level of 5–30 μL/mL (Liu et al., Proc Natl Acad Sci U S A, 2012, 109:12248-53; Obata et al., Clin Endocrinol, 2013, 79: 204-10; Benedetti et al., Hum Psychopharmacol, 2021, 36: e2793; Mavilia et al., J Dig Dis, 2021, 22: 214-21).

Your article is now acceptable for publication provided that you make the necessary corrections to the adiponectin serum units.

Reviewers' comments:

Reviewer's Responses to Questions

**Comments to the Author**

1. If the authors have adequately addressed your comments raised in a previous round of review and you feel that this manuscript is now acceptable for publication, you may indicate that here to bypass the “Comments to the Author” section, enter your conflict of interest statement in the “Confidential to Editor” section, and submit your "Accept" recommendation.

Reviewer #2: (No Response)

2. Is the manuscript technically sound, and do the data support the conclusions?

Reviewer #2: Yes

3. Has the statistical analysis been performed appropriately and rigorously? 

Reviewer #2: I Don't Know

4. Have the authors made all data underlying the findings in their manuscript fully available?

Reviewer #2: Yes

5. Is the manuscript presented in an intelligible fashion and written in standard English?

Reviewer #2: Yes

6. Review Comments to the Author

**Reviewer #2: **I appreciate the effort made by the authors in addressing all my last comments. However, there is one mistake that the authors should correct. Although they properly addressed my comments on IL-8 serum levels in the previous revised manuscript, I do not understand why the authors have also changed without reason the unit for serum adiponectin levels. Adiponectin is in fact the most abundant adipokine in human blood with a physiological level of 5–30 μL/mL (Liu et al., Proc Natl Acad Sci U S A, 2012, 109:12248-53; Obata et al., Clin Endocrinol, 2013, 79: 204-10; Benedetti et al., Hum Psychopharmacol, 2021, 36: e2793; Mavilia et al., J Dig Dis, 2021, 22: 214-21).

7. PLOS authors have the option to publish the peer review history of their article (what does this mean?). If published, this will include your full peer review and any attached files.

Reviewer #2: No

---

## [Editor Report · Acceptance letter]

10 Nov 2022

PONE-D-21-41053R2 

Altered serum adiponectin and interleukin-8 levels are associated in the pathophysiology of major depressive disorder: A case-control study 

Dear Dr. Islam:

I'm pleased to inform you that your manuscript has been deemed suitable for publication in PLOS ONE. Congratulations! Your manuscript is now with our production department. 

Kind regards, 

on behalf of

Dr. Sylvie Fournel-Gigleux 

Academic Editor

PLOS ONE